# OpenReview forum: "Inverse Scaling: When Bigger Isn't Better"
_TMLR — Accepted by TMLR_

### Review · Reviewer_bxfF · 2023-07-21

**Summary Of Contributions:**

The paper describes, presents, and characterizes the results from the Inverse Scaling Prize challenge. There were 11 prize winning datasets in the challenge, each of which induces a distinct instance of an inverse scaling law, which can manifest as U- and inverted-U-shaped scaling trends. The paper groups these datasets and other existing ones from the literature under a taxonomy of the kinds of weaknesses that can appear in larger LLMs that lead to this inverse scaling. Comprehensive experimental results are presented across a range of key model families. The paper discusses potential practical implications of the results, along with some possible remedies. The main overall effect of the paper is to provide inverse scaling with a comprehensive foundation, so that it can be more clearly understood and studied in the future.

**Audience:**

Yes

**Broader Impact Concerns:**

It's possible that the overall takeaways and specific weaknesses of LLMs identified in the paper could be used by bad actors looking to develop attacks on these kinds of models. At this point the beneficial contributions of the paper far outweigh the risks, but this risk would be worth mentioning.

**Claims And Evidence:**

Yes

**Requested Changes:**

As is, the paper is solid enough to recommend for acceptance, since the paper clearly validates the four potential causes of inverse scaling that are identified. However, the following changes would strengthen the work:
- Clarify in description of Classification Accuracy that the normalization happens over the probs for all classes, not all possible sequences. As written, it’s not immediately clear.
- Add discussion of limitations arising from underspecification aside from the mention in Pattern Match Suppresssion, in particular, limitations arising from non-apples-to-apples comparison between LLMs and humans in classification tasks.
- If relevant, address question above about how general higher model confidence relates to the SEQUENCE PROB metric.
- If possible, for more complete interpretability of results, provide a results plot for Prompt Injecting with non-log probabilities (at least in supplemental material; possible normalized in some way, e.g., based on model-specific expected confidence).
- Do you have any idea of a possible cause of the reversal w.r.t. K in the few-shot learning experiments? If so, it would be worth including.

Typos:
- In Section 1: “theimodel” -> “the model”
- In Section 3.1.1: “accuracy on its smallest model than on its largest model” -> “accuracy on its largest model than on its smallest model”

**Strengths And Weaknesses:**

Strengths:
- Provides a useful taxonomy of the types of failures that can occur. Uses the taxonomy and a consistent perspective to discuss each task.
- Carefully introduces a new benchmark that should be very useful for future developers and practitioners.
- Comprehensive evaluation gives a convincing picture of the state of the phenomenon.
- All in all, a compelling demonstration of how to use a competition to discover important behavioral dynamics in AI models. This may become a more and more important kind of paper as human labour is increasingly useful to investigate AI behavior.

Weaknesses:
- For some tasks, the comparison to Surge AI humans is not apples-to-apples. In particular, the LLMs do not know that they are choosing from a fixed set of classes for classification tasks, while the humans presumably do. For example, in the Memo Trap example of completing a quote, a more capable LLM may be planning a multi-sentence quote, but gets scored based on the completion of the first sentence. This is an example of a limitation arising from "underspecification" of the task. "underspecification" is mentioned in the discussion of Pattern Match Suppression, but it would be helpful to mention it for other tasks for which it applies, or add a general discussion of it somewhere else in the paper.
- It is difficult to immediately interpret the Prompt Injection results plot. Is it possible to plot the probabilities directly (perhaps normalized in some way based on the model?). Is it possible that some of the effect is due to the expected entropy in a model's predictions? E.g., if larger models are generally more confident in their wrong answers they will have higher loss than small models even if small models also get the answer wrong. If so, is there a way to this disentangle effect? Or would general higher confidence in wrong answers fall under "strong prior"?

---

> ### Author Response · Authors · 2023-09-02
> **Response to Reviewer bxfF**
>
> Thanks for your review. We are glad you found the taxonomy useful and our evaluation comprehensive. Addressing your feedback:
> - “For some tasks, the comparison to Surge AI humans is not apples-to-apples. In particular, the LLMs do not know that they are choosing from a fixed set of classes for classification tasks, while the humans presumably do.”
>   - This is a good point. Please see [this comment](https://openreview.net/forum?id=DwgRm72GQF&noteId=FHey08OelR) that we’ve posted separately as multiple reviewers asked about it.
> - It is difficult to immediately interpret the Prompt Injection results plot. Is it possible to plot the probabilities directly (perhaps normalized in some way based on the model?). Is it possible that some of the effect is due to the expected entropy in a model's predictions? E.g., if larger models are generally more confident in their wrong answers they will have higher loss than small models even if small models also get the answer wrong. If so, is there a way to this disentangle effect? Or would general higher confidence in wrong answers fall under "strong prior"?
>   - Overconfidence on incorrect answers is one mechanism through which inverse scaling could occur, so it is possible that some of the effect is due to expected entropy. However, we believe that increased confidence on incorrect answers is also a phenomenon that can be categorized as an adverse effect of scaling even if the classification accuracy remains the same
> - “Clarify in description of Classification Accuracy that the normalization happens over the probs for all classes, not all possible sequences.”
>   - We will make this more clear.
>   - Intended edit (addition in bold):
>     - “This metric can be used for standard classification tasks, for example when testing how well a model can choose the correct response. Each class could consist of multiple tokens, so we used the probability of the full token sequences (renormalized to sum to 1) to compute the classification loss, by evaluating the average negative log-probability of the correct response. **Normalization is performed over the multi-choice options given, rather than over all sequences.**”
> - “Add discussion of limitations arising from underspecification aside from the mention in Pattern Match Suppresssion, in particular, limitations arising from non-apples-to-apples comparison between LLMs and humans in classification tasks.”
>   - We can mention and explain the purpose of the human comparison (as described above) where appropriate.
> - “If relevant, address question above about how general higher model confidence relates to the SEQUENCE PROB metric.”
>   - We can add further discussion of model confidence.
> - “If possible, for more complete interpretability of results, provide a results plot for Prompt Injecting with non-log probabilities (at least in supplemental material; possible normalized in some way, e.g., based on model-specific expected confidence).”
>   -  We can add a plot of direct average probabilities on the correct token.
> - “Do you have any idea of a possible cause of the reversal w.r.t. K in the few-shot learning experiments? If so, it would be worth including.”
>   - No. This is an important question for future work.
> - “in the Memo Trap example of completing a quote, a more capable LLM may be planning a multi-sentence quote”
>   - This is a good point, and we can add it as a limitation of the task.
>   - Intended addition:
>     - “One limitation of this task is that LMs could be intending a multi-word completion which starts with the memorized word, but follows it with a novel ending. An improved version of this task might add a period to the end of the options to indicate that the sentence ends after that word.”

---

> > ### Comment · Reviewer_bxfF · 2023-09-05
> >
> > Thanks for addressing these questions/concerns. The clarifications provided make sense, and will be particularly useful in clarifying key directions for follow-up work. Thanks!

---

### Review · Reviewer_jSBV · 2023-07-30

**Summary Of Contributions:**

This paper presents and discusses the result of a public contest organized by the authors. The goal of the contest was to find tasks that large language models (LLMs) struggle with. More specifically, the authors tasked participants to find and submit tasks for which LLMs do not obey usual scaling laws, i.e. find tasks for which performance does not monotonically improve -- or outright worsens -- as more data and/or compute are used (as measured by training FLOPS). The authors claim that identifying these difficult tasks which obey **inverse** scaling is important to further improve LLMs, as these are the tasks that will not be solved just by throwing more compute at the problem or by collecting even larger datasets. To incentivize participants, the authors offered various prizes based on different criteria, ranging from \\$5,000 for third prizes, \\$20,000 for second prizes, and \\$100,000 for a first price. The authors awarded 11 third prizes, and no second nor first price as no submission satisfied all their criteria. In the paper, the authors carefully describe each of the submissions which were awarded a third prize, and hypothesize about the causes for the observed inverse scaling of LLMs at the submitted tasks.

**Audience:**

Yes

**Broader Impact Concerns:**

I have no immediate broader impact concerns.

**Claims And Evidence:**

Yes

**Requested Changes:**

Please discuss the points I raised in the "weaknesses" section of my review.

**Strengths And Weaknesses:**

Before the main body of my review, I wanted to highlight that I am not an expert in natural language processing (NLP) and LLMs, and I am not intimately familiar with the corresponding literature. While this paper is not really technical and can easily be understood by a non-expert in NLP, I do find framing it in the literature difficult. I believe this works shows convincing evidence that all the considered tasks indeed obey inverse scaling. I also believe that identifying these tasks is highly relevant. However, I do not know the level of awareness that NLP researchers already have about these failure modes, nor how formally these tasks had previously been rigorously established as failures modes of LLMs. Thus, while I believe this paper will be highly interesting to a large audience of NLP researchers, I do have a relatively high amount of uncertainty about it (due to my lack of expertise, not to any concerns with the paper).

### Strengths ###

- The paper is very well written and easy to follow.
- The paper presents clear evidence of inverse scaling across a large number of LLMs for the presented tasks.
- I agree with the authors that identifying failure modes of current LLMs which cannot be addressed just with more compute/data is a valuable way of guiding future research.
- The authors are quite thorough in their analysis, proposing hypotheses for why each of the tasks scales the way it does.

### Weaknesses ###

While I cannot think of any glaring weakness, I do think the discussion around some tasks is not detailed enough, or can seem slightly self-contradictory. In particular:

- For the modus tollens task (3.2.1), the authors speculate that LLMs fail because humans are bad at this task, and thus minimizing the training objective implies imitating human behaviour. Yet, Table 1 shows humans are very good at the modus tollens task (which I find surprising, and would appreciate if the authors could further clarify).

- Similar to modus tollens, could it be that for some of the distractor task tasks, humans are also bad, and this is what causes LLMs to not do well? In particular, for the NeQA task (3.3.3), I could imagine a non-negligible number of humans read the sentence too quickly and miss the word "not", causing them to answer incorrectly (although this is not reflected in Table 1, similar to modus tollens). Similarly, for the Sig Figs task (3.3.4), I would also have the intuition that a non-negligible percentage of humans do not really know the meaning of significant digits, and thus fail at the task as well. Could you please elaborate on these points?

- I think a more detailed discussion about how the human data was collected is warranted. Previous work has shown that human subjects from crowdsourcing platforms can cheat and use LLMs [1]. Clearly this did not happen here, as the task was tested on the same LLM that the humans could have used for cheating. Was there any measure preventing cheating? I do find some of the numbers in Table 1 to be surprisingly high (i.e. I'd expect a bit more human error, could you please discuss this?).

- The discussion about BIG-Bench in 4.1 is a bit short, and does not highlight the difference between previous work and this work.


Finally, some typos/formatting issues:
- The first page has a huge amount of white space. If this is simply because the author list is very long and will occupy the entirety of the page then this is not an issue, but if this is not the case, please address this.
- Page 2: "theimodel"
- Some boxes containing "Input:" and "Output:" align the "I" and "O", respectively. I think it would be slightly easier to read if the ":" symbols were aligned instead, as this would make it visually easier to spot the difference (e.g. in 3.1.1)
- 4.2: missing parenthesis

[1] Artificial Artificial Artificial Intelligence: Crowd Workers Widely Use Large Language Models for Text Production Tasks, Vaselovsky et al., 2023

---

> ### Author Response · Authors · 2023-09-02
> **Response to Reviewer jSBV**
>
> Thank you for your review. We are glad you found the evidence clear and the analysis thorough. Addressing your points:
> - “ [...] human subjects from crowdsourcing platforms can cheat and use LLMs [...] Was there any measure preventing cheating? I do find some of the numbers in Table 1 to be surprisingly high (i.e. I'd expect a bit more human error, could you please discuss this?).”
>   - This is a good point. Please see [this comment](https://openreview.net/forum?id=DwgRm72GQF&noteId=FHey08OelR) that we’ve posted separately as multiple reviewers asked about it.
> In particular, cheating is not an issue since contractors were specifically allowed to use outside assistance to help them get to the correct answer.
> Additionally, data collection was done before the release of ChatGPT so LLM assistance was probably much less widespread.
> - “[...] the authors speculate that LLMs fail because humans are bad at this task, and thus minimizing the training objective implies imitating human behaviour. Yet, Table 1 shows humans are very good at the modus tollens task [...]”
>   - Our hypothesis here is that the internet contains many examples of poor modus tollens reasoning from humans (one extreme example of this is “modus schmollens”, where ‘reasoners conclude that “the soup tastes like garlic” from the premises “If a soup tastes like garlic, then there is garlic in the soup; Carole tells Didier that there is no garlic in the soup they are eating”), often in contexts not explicitly related to formal logic. We believe this is not inconsistent with good performance by human contractors who are being asked specifically about the validity of a modus tollens inference and given permission to use external resources.
> - “Similar to modus tollens, could it be that for some of the distractor task tasks, humans are also bad, and this is what causes LLMs to not do well? In particular, for the NeQA task (3.3.3), I could imagine a non-negligible number of humans read the sentence too quickly and miss the word "not", causing them to answer incorrectly (although this is not reflected in Table 1, similar to modus tollens).”
>   - Similarly to modus tollens, since we are directly asking the contractors to answer the questions, we expect that they may be more careful to read them correctly than the average internet discussion.
> - “Similarly, for the Sig Figs task (3.3.4), I would also have the intuition that a non-negligible percentage of humans do not really know the meaning of significant digits, and thus fail at the task as well.”
>   - This seems plausible, and as discussed above, our contractors were permitted to consult the internet for help on the definition of significant digits.
> - “The discussion about BIG-Bench in 4.1 is a bit short, and does not highlight the difference between previous work and this work.”
>   - We can add further discussion on how BIG-Bench differs from this work, namely the goal of BIG Bench was to crowdsource LM evaluations in general rather than ones specifically targeted to reveal inverse scaling behavior in LMs.
>   - Intended change (addition in bold):
>     - “BIG-Bench \citep{bigbench} is a large collection of more than 200 tasks sourced from the LM research community.  As discussed in \S \ref{sec:social_bias} below, BIG-Bench contains some tasks that demonstrate inverse scaling, but does not actively solicit inverse scaling tasks, and does not discuss potential causes. Some BIG-Bench tasks also show U-shaped scaling, as discussed in \citet{u_shaped}.”

---

> > ### Comment · Reviewer_jSBV · 2023-09-03
> > **Response to response**
> >
> > Thank you for your reply, which does clarify the small concerns I had. I have also read the other reviews and responses, and have no additional concerns.

---

### Review · Reviewer_FLAq · 2023-08-21

**Summary Of Contributions:**

This paper discusses a public contest that was held to explore the inverse scaling of large language models (LLMs). The goal of the contest was to identify tasks where performance decreased as model scale increased. In the past year, the contest received 99 submissions, and 11 were awarded for successfully demonstrating inverse scaling. Based on these submissions, the paper identifies several potential causes of inverse scaling and includes discussions on the possibility of U-shaped scaling.

**Audience:**

Yes

**Broader Impact Concerns:**

No concerns.

**Claims And Evidence:**

Yes

**Requested Changes:**

1. The use of FLOPS as the sole metric for measuring scaling may not accurately capture the inverse scaling effect. It would be helpful to provide a more detailed explanation of why this is the case.

2. Given that models such as GPT-2/3, OPT, and Gopher are trained with very different data and training setups, it may be more informative to study the scaling law for models within the same family, such as OPT 125M/350M/1.3B/2.7B/13B/30B/66B/175B. Please add descriptions on the advantages and disadvantages of using models from different families versus your current approach for studying inverse scaling.

3. The reviewer acknowledges that running an inverse contest is helpful, but it would be beneficial to better characterize the technical novelty of this paper.

**Strengths And Weaknesses:**

Strengths:

1. The contest contributes to a better understanding of LLMs as their sizes are scaled, making it possible to improve the pre-training process or develop mitigation strategies during fine-tuning to make LLMs more widely useful.

2. Considerable thought has been put into selecting inverse scaling tasks and designing tests to attract participants and provide useful information.

3. The paper presents a collection of potential causes for inverse scaling behaviors, providing an initial characterization of inverse scaling.

Weaknesses:

1. While the reviewer likes the idea of running an inverse scaling contest, the methodology used for this study is a bit concerning. For instance, the scaling is measured solely in terms of FLOPS, which may not be an accurate or convincing metric for measuring the scaling effect. The paper argues that FLOPS is a better proxy and cites the Chinchilla paper by Hoffmann et al. However, the Chinchilla paper actually argues against using a single metric to measure the scaling effect and advocates for considering both model size and the number of tokens. For example, both Figure 1 and Table 1 in the Chinchilla paper measure the scaling effect in terms of both model size and the number of training tokens. And using purely FLOPS may not lead to compute-optimal LLMs, e.g., a very large model trained on a relatively small number of tokens. Therefore, using only the FLOPS metric to study inverse scaling may be flawed and hurts the soundness of the conclusion.

2. While the potential causes identified are interesting, it is unclear whether they are the original findings of this paper or simply a summary of findings from the contest submissions. If it is the latter, it raises questions about the technical contributions of this paper. Does the paper only summarize other people's findings?  If the common causes were indeed discovered by the authors, then credit should be given to this paper.

3. The manuscript appears to primarily describe how a public contest was run to find tasks that exhibit an inverse correlation with pretrained LM optimization objectives. Much of the content is about how the Inverse Scaling contest and tasks were set up. As such, it does not seem to be a typical technical paper and reads more like a mixture of contest announcements and its result summarization. The drawback of this way of writing is that it makes it difficult to see the technical novelty of the paper, which is crucial for acceptance.

---

> ### Author Response · Authors · 2023-09-02
> **Response to Reviewer FLAq**
>
> Thank you for your review. We are glad you liked the idea of the Prize and found the potential causes interesting. Addressing your points:
> - “the scaling is measured solely in terms of FLOPS, which may not be an accurate or convincing metric for measuring the scaling effect” and “[since models] are trained with very different data and training setups, it may be more informative to study the scaling law for models within the same family”
>   - We believe there is no issue with the way we use FLOP comparisons in our paper, since our key FLOP comparisons are within individual model families (each line on the graph represents a model family). Are you claiming that FLOPs aren't a good proxy for scale within the Gopher or Anthropic model series?
>   - FLOPs work well as a measure within a model family, since they typically either keep the dataset size fixed, or increase it as the model size increases.
>   - We will clarify in our paper that readers should not make FLOP comparisons across models from different model families, due to the issues you have mentioned and others.
> - “it is unclear whether [the potential causes] are the original findings of this paper or simply a summary of findings from the contest submissions”
>   - The potential causes were proposed by the primary authors of the paper, since all contest winners whose work is featured in the paper were invited to be co-authors of the paper.
> - “it would be beneficial to better characterize the technical novelty of this paper.”
>   - To summarize our contributions:
>     - We discovered 11 cases of inverse scaling in LMs.
>     - We analyzed those submissions and identified possible causes of inverse scaling behavior, serving as the first systematic examination and explanation of the phenomenon of inverse scaling
>     - We collected instances of inverse scaling in the literature and show how they fit into our explanation of why inverse scaling happens in LMs.
>   - We will revise our paper to reflect the technical novelty of our work.

---

> > ### Comment · Reviewer_FLAq · 2023-09-05
> > **Response to the rebuttal**
> >
> > I have read the authors' responses and my concerns have been addressed. It would be helpful to highlight the clarification provided in this response within the paper itself. For example, it should be emphasized that FLOP comparisons are only made within individual model families, as comparing FLOPs across different model families is not particularly meaningful.

---

### Comment · Reviewer_bxfF · 2023-09-01
**Addressing clarification questions?**

All reviewers agree that the Claims and Evidence are generally correct, but each raised some specific issues around clarifying some details of the claims. Are the authors planning to address any/all of these with revisions?

---

> ### Author Response · Authors · 2023-09-02
> **Responses added**
>
> Thank you for the reminder. We have posted responses to all of the reviewers with our intended clarifications.

---

### Author Response · Authors · 2023-09-02
**General comment on human contractor scores**

A general point regarding all of the concerns with the human scores is that these were intended to validate that the task labels given by the task author were correct, rather than to act as the performance an average human would receive. While the comparison is not exactly one-to-one (since for example, contractors could see the multiple-choice options while models could not), this does not undermine the scores since the goal is to verify whether humans would agree that the provided task labels are correct, rather than asses their ability to perform the task. This is because it is easy to construct an inverse scaling task if the answers were allowed to be incorrect: a dataset of simple True/False questions with the True and False labels switched would show inverse scaling. Thus contractors were allowed to use the internet to help them come to a correct answer to the question. We’ll clarify in the paper.
- Intended addition:
  - “The purpose of the human agreement scores is to ensure that the labels given by task authors represent the answer that humans would agree, upon reflection, was correct for that task. This is because it is easy to construct an inverse scaling task if the answers were allowed to be incorrect: a dataset of simple True/False questions with the True and False labels switched would show inverse scaling. Contractors were allowed to use the internet to help them come to a final answer to each question, since this meant  it was less likely that mistakes or a lack of knowledge would affect the scores.”

---

### Decision · Action_Editors · 2023-09-12

**Recommendation:** Accept with minor revision

**Comment:**

There is a unanimous decision from the reviewers, who think the paper is worth publishing. All reviewers think the paper is interesting and timely to study an important property of LLM. There are some concerns from the reviewers as well, which I believe have been clarified during the rebuttal process. The authors promise to add more evidences and clarifications in the revisions, e.g., add a plot of direct average probabilities on the correct token. I request the authors to follow the reviewers' comments to revise the paper accordingly.

Finally, I do like the work the authors have done, and the presentation of the paper. Given that the paper presents an important issue of the emerging LLM topic, and the supports from the reviewers, I recommend the paper to be Featured Certificated.

**Audience:**

Studying properties of LLMs is an emerging research areas, this paper will benefit the increasingly large LLM community.

**Claims And Evidence:**

This paper describes a public contest to explore the so-called inverse scaling of large language models (LLMs), and presents several cases of the cases that indeed follow the inverse scaling law, and discusses some potential reasons. Based on a large-scale of evaluations submitted in the contest, the authors make some claims which are supported by extensive experimental evidence. According to the reviews, the claims are accurate and evidences are solid.